# Recurrence of Hepatic Encephalopathy after TIPS: Effective Prophylaxis with Combination of Lactulose and Rifaximin

**DOI:** 10.3390/jcm10204763

**Published:** 2021-10-17

**Authors:** Leon Louis Seifert, Philipp Schindler, Martin Schoster, Jan Frederic Weller, Christian Wilms, Hartmut H. Schmidt, Miriam Maschmeier, Max Masthoff, Michael Köhler, Hauke Heinzow, Moritz Wildgruber

**Affiliations:** 1Medical Clinic B, Department of Gastroenterology, Hepatology, Endocrinology, Infectiology, University Hospital Muenster, 48149 Muenster, Germany; leonlouis.seifert@ukmuenster.de (L.L.S.); martin.schoster@ukmuenster.de (M.S.); christian.wilms@ukmuenster.de (C.W.); hartmut.schmidt@uk-essen.de (H.H.S.); m.maschmeier@bk-trier.de (M.M.); h.heinzow@bk-trier.de (H.H.); 2Clinic for Radiology, University Hospital Muenster, 48149 Muenster, Germany; philipp.schindler@ukmuenster.de (P.S.); Max.Masthoff@ukmuenster.de (M.M.); michael.koehler@ukmuenster.de (M.K.); 3Department of Hematology, University Hospital Tuebingen, 72076 Tuebingen, Germany; Jan.Weller@med.uni-tuebingen.de; 4Department of Gastroenterology and Hepatology, University Hospital Duisburg-Essen, 45147 Essen, Germany; 5Department of Medicine I, Krankenhaus der Barmherzigen Brüder, 54292 Trier, Germany; 6Department of Radiology, University Hospital LMU Munich, 81377 Munich, Germany

**Keywords:** transjugular intrahepatic portosystemic shunt, decompensated liver cirrhosis, complications of liver cirrhosis, portal hypertension, hepatic encephalopathy

## Abstract

Background: Transjugular intrahepatic portosystemic shunt (TIPS) implantation is an established procedure to treat portal hypertension with hepatic encephalopathy (HE) as a common complication. There is lack of evidence concerning HE prophylaxis after TIPS. Methods: N = 233 patients receiving TIPS between 2011 and 2018 at a German tertiary care center were included. Of them, 21% (n = 49) had a history of HE. The follow-up period was 12 months. The risk factors of post-TIPS HE were analyzed via multivariate analysis. The efficacy of prophylactic medication regimens was studied. The results show that 35.6% (n = 83) received no medication (*NM*), 36.5% (n = 85) received lactulose monoprophylaxis (*LM*), 2.6% (n = 6) rifaximin monoprophylaxis (*RM*) and 25.3% (n = 59) lactulose and rifaximin (*LR*) of which 64.4% received l-ornithin-l-aspartate (*LOLA*) additionally (*LR + LOLA*) and 36.6% did not (*LRonly*). Results: Multivariate analysis revealed higher age (*p* = 0.003) and HE episodes prior to TIPS (*p* = 0.004) as risk factors for HE after TIPS. *LM* has no prophylactic effect. *LR* prevents HE recurrence at 1, 3 and 12 months after TIPS (*p* = 0.003, *p* = 0.003, *p* = 0.006) but does not prevent HE in patients with no history of HE (*p* = 0.234, *p* = 0.483, *p* = 0.121). *LR* prevents HE recurrence compared with *LM/NM* (25.0% vs. 64.7%, *p* = 0.007) within 12 months after TIPS, whereas de novo occurrence is unaffected (*p* = 0.098). The additional administration of *LOLA* to *LR* has no benefit (*LRonly*: 25.0%, *LR + LOLA*: 29.7%, *p* = 0.780). Conclusions: Higher age and previous HE are risk factors post-TIPS HE. In patients with HE prior to TIPS, effective prophylaxis of HE is feasible via combination of lactulose and rifaximin with no additional benefit from *LOLA*.

## 1. Introduction

Transjugular intrahepatic portosystemic shunt (TIPS) implantation is used in the management of complications of portal hypertension [1]. The procedure is safe and improves transplant-free survival, both in patients with refractory ascitic decompensation and refractory or recurrent variceal bleeding [2,3]. Hepatic encephalopathy is a common complication of liver cirrhosis [4]. Several factors contribute to the development of HE in cirrhotic patients, such as splanchnic blood shunting, the intestinal overproduction of neurotoxins, impairment of the intestinal blood barrier and the reduction in the hepatic elimination of neurotoxins [4,5,6]. The manifestation of HE shows a broad range, from mild cognitive impairment to severe comatose states, classified according to the West–Haven Criteria [7,8]. After TIPS implantation, HE occurs in up to 50% of patients [9,10]. The basic treatment of HE includes the avoidance of a high protein diet and the application of enemas [11,12]. The pharmaceutical treatment of overt HE consists of the administration of non-absorbable disaccharides such as lactulose. In recurrent or refractory HE, besides the consideration of invasive TIPS modification, rifaximin, as a non-adsorbable antibiotic, can be added as a pharmaceutical treatment [7,13]. L-Ornithin-L-Aspartat (*LOLA*) reduces ammonia levels by the stimulation of urea synthesis and glutamine synthesis [14]. *LOLA* can be added to the treatment regimen and applied both orally and intravenously. Currently, a routine pharmacological prophylaxis of post-TIPS HE is not recommended, according to the guidelines of both the American Association for the Study of Liver Diseases (AASLD) and the European Association for the Study of the Liver (EASL), as evidence for effective prophylaxis is lacking [7]. There is a need for scientific evidence to develop common recommendations for the effective prophylaxis of HE after TIPS [15,16]. The aim of this study is to evaluate the efficacy of different pharmacological regimens in preventing the development of HE after TIPS.

## 2. Materials and Methods

### 2.1. Study Design and Data Collection

The primary endpoint of this retrospective study is the occurrence of HE within 1, 3 and 12 months from a TIPS implantation. The study was approved by the local ethics committee (2021-056f-S) and conducted in accordance with the Declaration of Helsinki. Informed consent was waived due to the retrospective character of the study.

All patients receiving a TIPS placement between 2011 and 2018 at a German tertiary care medical center were included in the analysis. Data from 344 patients was available. Only adult patients were included. Patients were only included if a TIPS was placed for treatment of portal hypertension in cirrhotic patients. Thus, TIPS indications were ascites refractory to pharmacological treatment, variceal bleeding (including secondary prophylaxis, a preemptive TIPS within 24–72 h of initial variceal bleeding or an emergency TIPS) or both ascites and history of variceal bleeding. We excluded patients in whom information concerning HE prophylaxis was not available. Patients after LTX were included in cases of the recurrence of cirrhosis and the presence of respective TIPS indications. Data from a total of 233 patients were included in the final analysis. Data were collected retrospectively through electronic record review. Laboratory data and clinical data were assessed for the three days preceding the day of their TIPS implantation. Follow-ups were performed at months 1, 3 and one year after their TIPS implantation. Follow-up data were collected for 12 months or until death or liver transplantation. HE was diagnosed according to the West–Haven criteria [8]. Due to the lack of universal criteria of minimal HE, only episodes with clinical features of HE were included (covert HE: grade I, overt HE: grade II–IV).

The pharmacological prophylaxis of HE was performed by the administration of lactulose, rifaximin or *LOLA*. Prophylactic medication was initiated within 72 h prior to TIPS implantation and for at least 12 months afterward. No data concerning prior ongoing prophylactic regimens prior to the TIPS evaluation was available. Lactulose was dosed individually with the aim of two-to-three loose stools per day. Rifaximin was administered twice daily at a dosage of 550 mg. *LOLA* was applied orally at a dosage of 3–6 g thrice per day. In case of occurrence of HE post TIPS implantation, all secondary prophylaxis was performed with a combination of lactulose and rifaximin, with or without LOLA.

A total of 35.6% of patients received no medication (*NM*) for HE prophylaxis, 36.5% of patients received lactulose monoprophylaxis (*LM*), 2.6% received rifaximin monoprophylaxis (*RM*) and 25.3% of patients received a combined prophylactic regimen consisting of lactulose and rifaximin (*LR*). Among these patients, 64.4% received *LOLA* (oral administration) additionally (*LR + LOLA*) and 36.6% did not (*LRonly*). A prophylactic regimen was chosen and applied according to the respective standard operating procedures and available guidelines, with variability throughout the observational period. Standard operating procedures were based on the guidelines of the EASL, the AASLD and the DGVS (German society for digestive and metabolic diseases) and complemented according to internal analyses. The addition of LOLA as part of a primary prophylactic regimen after TIPS implantation was part of an internal SOP and not based on international guidelines. 

### 2.2. TIPS Procedure

TIPS implantation was performed by experienced interventional radiologists according to standard operating procedures. Intrahepatic needle positioning was controlled via sonography to control access to the portal vein. Shunts were sized to 8-mm diameters, aiming for a portosystemic gradient < 12 mmHg. Only in cases of insufficient reduction of the portosystemic gradient (>12 mmHg after placement) was a further dilatation of the shunt diameter, to 9 and 10 mm, performed. 

### 2.3. Statistical Analysis

Statistical analysis was performed using *SPSS* version 26.0 (SPSS Inc., Chicago, IL, USA) and *R* version 3.5.3 (R Foundation for Statistical Computing, Vienna, Austria). All data are presented as the mean (standard deviation, SD), median (range), absolute or percentage, depending on the nature of variables and distribution. The Chi-square test was used for the contingency tables. The paired student’s *t*-test was used for quantitative data and the Mann–Whitney U test was used for qualitative data of non-normal distribution. Two-sided *p*-values < 0.05 were defined as statistically significant. For the analysis of risk factors for HE development after a TIPS implantation a multivariate Cox regression model was created. Multivariable Cox-regression analysis was performed using forward variable selection. Logistic regression analysis was used for analysis of prophylactic potency of the different prophylactic regimen in comparison patients to *NM*. Kaplan–Meier curves and the log-rank test were used to analyze the occurrence of HE. 

## 3. Results

### 3.1. Risk Factors for HE

Demographic and baseline characteristics of the entire cohort are presented in Table 1. We performed multivariate Cox regression analysis to identify predictors of developing HE after a TIPS implantation. Only patients with no prophylactic regimen for HE after their TIPS implantation were included in this analysis. All factors identified as significant risk factors (*p* < 0.05) in a univariate model were included in the multivariate analysis. Via multivariate analysis, higher patient age and a history of at least one HE episode prior to the implantation of a TIPS were identified as risk factors for developing HE afterward (see Table 2).

### 3.2. Prophylactic Regimens

The cohort was divided according to whether at least one episode of HE was observed prior to TIPS placement (21%, n = 49) or not (79%, n = 184). A total of 93.1% (46 patients) of patients with a history of HE episodes prior to a TIPS implantation had experienced HE episodes of grade I or II in the past, and only 6.1% (3 patients) had experienced grade III HE episodes and no patient had grade IV HE episodes. Of the three patients with grade III HE, two patients had refractory variceal bleeding and one patient had refractory ascites as a TIPS indication. All grade III HE episodes had occurred more than 12 weeks prior to a TIPS implantation. No symptoms were present at the time of the TIPS implantation. Patients who had experienced at least one episode of HE prior to their TIPS placement had significantly more severe cirrhosis, as depicted by higher MELD-scores and higher proportions of high Child–Pugh grades (*p* < 0.001 respectively). In patients with an episode of HE prior to a TIPS implantation, HE prophylaxis with *LR* was performed in 59.2% of patients (*NM* 16.3%, *LM* 20.4%, *RM* 4.1%) compared with 16.3% in patients with no HE episode prior to a TIPS placement (*NM* 40.8%, *LM* 40.8%, *RM* 1.6%). Baseline characteristics according to the regimen of HE prophylaxis are presented in Appendix A. Patients who did receive *LR* presented with significantly more severe cirrhosis according to MELD score and Child–Pugh stadium at the time of TIPS implantation (*p* < 0.001 respectively).

### 3.3. Logistic Regression Analysis of Prophylactic Efficacy

For the evaluation of the prophylactic potency of the different regimens, compared to *NM,* we performed logistic regression analyses (see Table 3). With regard to HE episodes prior to a TIPS implantation, HE prevention via *LR* was only effective in those who had experienced HE prior to a TIPS implantation and not in patients without previous HE episodes (HE prior TIPS: 1 month, OR = 0.048 *p* = 0.003; 3 months, OR = 0.042 *p* = 0.003; 12 months, OR = 0.056 *p* = 0.006; no HE prior TIPS: 1 month, OR = 0.490 *p* = 0.234; 3 months, OR = 0.647 *p* = 0.483; 12 months, OR = 0.450 *p* = 0.121). Both *LRonly* and *LR + LOLA* did not prevent de novo HE effectively in patients with no prior HE episodes (*LRonly*: 1 month, OR = 0.940 *p* = 1.000; 3 months, OR = 0.768 *p* = 0.759, 12 months, OR = 0.500 *p* = 0.358; *LR + LOLA*: 1 month, OR = 0.251 *p* = 0.119; 3 months, OR = 0.558 *p* = 0.397; 12 months, OR = 0.417 *p* = 0.177). In patients with HE prior to a TIPS implantation, on the other hand, *LRonly* and *LR + LOLA* both prevented HE recurrence effectively at all time points (*LRonly*: 1 month, OR = 0.057 *p* = 0.041; 3 months, OR = 0.024 *p* = 0.010, 12 months, OR = 0.0.24 *p* = 0.010; *LR + LOLA*: 1 month, OR = 0.044 *p* = 0.005; 3 months, OR = 0.051 *p* = 0.009; 12 months, OR = 0.071 *p* = 0.024).

### 3.4. LM Has No Prophylactic Potency

The efficacy of the different prophylactic regimen was further analyzed using the Kaplan–Meier method and log-rank test. The frequency of HE within 1 month, 3 months or 12 months did not differ significantly between patients who received *NM* (53.7% of patients with HE after a TIPS implantation within 12 months) or *LM* (50.6% of patients with HE after TIPS within 12 months) for HE prophylaxis as presented in Figure 1 (log-rank test, 1 month: *p* = 0.581; 3 months: *p* = 0.723; 12 months: *p* = 0.808). All patients who received *RM* developed HE after their TIPS was placed.

### 3.5. LR Prevents HE Compared to LM/NM

In all patients who received *LR* (lactulose and rifaximin, n = 59) HE occurred significantly less often compared to patients who received either *NM* or *LM* within 12 months (28.1% vs. 52.1%, *p* = 0.004, log-rank-test; see Figure 2). A significant difference was already observed at one month (36.1% and 15.8%, *p* = 0.007, log-rank test) and three months after their TIPS implantation (42.6% and 25.0%, *p* < 0.05, log-rank test). 

### 3.6. LR Prevents HE Recurrence and Not De Novo Occurrence Post TIPS Implantation

The protective effect of the combined prophylactic regimen was confirmed when including only patients who had experienced at least one HE episode prior a TIPS placement into analysis. HE frequency within after 1, 3 and 12 months after their TIPS implantation was 10.7%, 20.0% and 25.0% if patients received *LR* and 44.4%, 61.1% and 64.7% in patients with *NM*/*LM* (log-rank test, *p* = 0.009, *p* = 0.003, *p* = 0.007; see Figure 3a). No significant effect was found in patients without a HE-episode prior to a TIPS implantation at any of the respective time points (*LR*: 20.7%, 29.6% and 31.0% after 1 month, 3 months and 12 months; *NM/LM*: 33.7%, 39.6% and 50.7%; log-rank test *p* = 0.173, *p* = 0.364, *p* = 0.098; see Figure 3b).

### 3.7. LOLA Provides No Additional Prophylactic Effect

No significant differences were found between patients receiving either *LRonly* (25% HE post TIPS placement within 12 months) or *LR + LOLA* within 1, 3 or 12 months (29.7% patients with HE post TIPS placement within 12 months) (*p* = 0.780, log-rank test, see Figure 4), irrespective of prior HE episodes. The prophylactic efficacy of *LR* was not significantly affected or improved by additional administration of *LOLA* (*p* = 0.780, log-rank test; see supplementary Appendix A).

### 3.8. Development of Chronic HE

Developing chronic HE post TIPS implantation is refractory to pharmacological treatment, resulting in the necessity of a TIPS reduction or occlusion, observed in nine patients. Among these patients, one patient had a history of HE prior to their TIPS placement. TIPS revision became necessary in seven patients (*NM* two patients, *LM* three patients, *RM* one patient, *CPonly* one patient) after TIPS implantation due to chronic HE after TIPS implantation. Chronic post-TIPS HE was treated with TIPS occlusion in two patients (*NM* one patient, *CPonly* one patient).

## 4. Discussion

Hepatic encephalopathy remains a challenging complication after TIPS insertion [13]. With further emergence of TIPS implantations in the management of cirrhotic patients with portal hypertension, the lack of evidence concerning HE prophylaxis has become clear [17,18]. The results of this retrospective analysis support the use of a combined prophylactic medication regimen with lactulose and rifaximin to prevent the recurrence of HE after a TIPS implantation. Supplementary administration of oral LOLA cannot be recommended.

Development of HE is a severe complication in patients with liver cirrhosis, having a severe impact on patient morbidity and mortality [19]. With each HE episode patient prognosis deteriorates [20]. Nonetheless, the primary prophylaxis of HE with lactulose is not recommended in cirrhotic patients except for conditions with precipitating factors, such as gastrointestinal bleeding [21,22]. Interestingly, rifaximin has been proven to be as effective in that situation as lactulose [23]. Secondary prophylaxis, on the other hand, should be administered and current guidelines recommend the use of lactulose (individual dosage with the aim of two to three loose stoles per day) [7,15,24]. After a second episode of HE, rifaximin can be added to the secondary prophylactic regimen [7]. There is emerging evidence that adding rifaximin improves the (secondary) prophylactic potency with an excellent safety profile [25].

With *LOLA* being well-established in the treatment of HE—especially in more severe cirrhosis—its role in the prophylaxis of HE remains disputed [26]. It is important to outline that the oral administration of *LOLA* is discussed controversially. Randomized clinical trials have resulted in contradictory findings in cirrhotic patients without a TIPS and a history of HE [27,28]. At least in one randomized controlled trial, the oral administration of *LOLA* was effective in preventing HE episodes within 6 months after randomization as compared to placebo [28]. In a randomized clinical trial, Bai et al. were able to show that the prophylactic administration of *LOLA* reduced serum fasting ammonia levels and improved the performance of patients in psychometric testing. However, the incidence of post-TIPS HE was not reduced significantly in this small study (total n = 40) [29]. In that study, *LOLA* was administered intravenously for 7 consecutive days after a TIPS implantation.

The recommendation not to routinely apply prophylactic medication for preventing HE after a TIPS implantation owes to a lack of evidence of a beneficial effect [7]. A prospective study has shown that prophylactic monotherapy with either lactulose or rifaximin is insufficient to effectively prevent post-TIPS HE [30]. However, the question if and in whom to administer a prophylactic regimen remains controversial and further evidence is required [16]. Other unanswered questions include the ideal timing and the duration of prophylactic treatment, since existing guidelines remain imprecise on this subject. Recently, the PEARL trial was proposed as a prospective multicenter randomized, double blind, placebo-controlled trial to investigate the impact of HE prophylaxis with lactulose and rifaximin in TIPS patients. According to the protocol, patients will receive lactulose and rifaximin from 72 h prior to a TIPS implantation and for 3 months afterwards [31]. With an estimated recruiting period of 3 years from today for the PEARL trial, the presented study outlines the preceding evidence to support further prospective studies and demonstrate the effective feasibility of HE prophylaxis with lactulose and rifaximin.

As a limitation, we observed significant differences in the baseline characteristics of patients receiving the different prophylactic regimens (or none; see Table 1 and Appendix A). In patients with a history of HE episodes prior to a TIPS implantation HE prophylaxis was more effectively achieved in the group receiving combined prophylaxis. Importantly, patients receiving the combined prophylaxis had significantly more severe liver cirrhosis, as indicated by their Child–Pugh scores and MELD scores, which contributed to a higher risk of HE developing post-TIPS implantation. These observed differences in the baseline characteristics of patients underline the prophylactic potency of the combination of lactulose and rifaximin. Due to the limited number of patients herein, a further analysis in a matched cohort was not feasible. 

Recent studies have focused on risk stratification to improve patient selection and improve patient safety [32]. There are also new tools to implement easy and fast risk stratification for HE after TIPS [33]. Several factors have been shown to contribute to development of HE after TIPS implantation, including the severity of cirrhosis, the etiology of liver disease, a portosystemic pressure gradient, higher age, the presence of diabetes and a low sodium score [13,33,34]. Our findings show that patients with a history of at least one HE episode prior to their TIPS implantation are at higher risk to develop HE afterward. A careful assessment with a distinct focus on HE episodes should just be considered pivotal in patient selection for an elective TIPS implantation. Interestingly, more severe cirrhosis was not identified as a risk factor for development of post-TIPS HE. 

A distinct patient selection with regard to HE history is not always feassible in a real-life setting. Salvage or preemptive TIPS implantation is necessary in a subgroup of patients with acute or refractory variceal bleeding [3]. Even though risk of HE should be considered in these patients as well, treatment of (sub-) acute bleeding should be at the center of decision-making concerning TIPS indication. Additional HE prophylaxis via systemic antibiotic treatment should be performed in these patients. Yet, gastrointestinal bleeding is only one of several precipitating factors (severe infection, spontaneous bacterial peritonitis, obstipation and others) for the development of HE [35]. It is furthermore advisable to distinguish between HE episodes with and without precipitating factors to estimate the risk of HE recurrence [7]. In our study, a clear distinction was not feasible on account of its retrospective performance. Future studies should take potential precipitating factors into consideration for the further improvement of risk stratification.

Recent studies have recently focused on the impact of sarcopenia on the risk of post-TIPS HE development. A prospective study identified the presence of sarcopenia as a predictor of post-TIPS HE [36]. Benmassaoud et al. later confirmed these findings in a retrospective analysis, including only patients with refractory ascites as TIPS indications but did not find sarcopenia to be a predictor of post-TIPS implantation survival [37]. Due to lacking data concerning presence of sarcopenia a possible bias concerning the presented data on post-TIPS HE development cannot be excluded.

Other limitations have further to be acknowledged. Data concerning ongoing HE prophylaxis prior to TIPS implantation or evaluation is not available. Also, the relatively small number of patients has to be taken into account. Due to the retrospective character of our study, the immediate transferability to patient care is limited and the results have to be considered with caution. Retrospectively, a patient selection bias cannot be excluded. Adherence to the prescribed prophylactic regimens was only controlled at follow up visits at 3, 6 and 12 months after TIPS implantation. Additionally, diagnosis of HE is difficult and prone to interindividual differences despite common guidelines. 

## 5. Conclusions

The presented findings imply important information to further improve patient management after TIPS. The administration of a prophylactic regimen with lactulose and rifaximin was shown to be effective to prevent HE after a TIPS implantation in patients with a history of HE episode prior to their TIPS implantation. Intensification of prophylactic medication by oral administration of LOLA showed no additional benefit in these patients. Prospective studies are needed to further improve risk stratification and identify patients who may benefit most from an intensified prophylactic regimen. With prospective studies being proposed and recruited for, lactulose and rifaximin can be used for prophylaxis of HE in patients with high risk for post-TIPS HE until data of prospective trials becomes available.

## Figures and Tables

**Figure 1 jcm-10-04763-f001:**
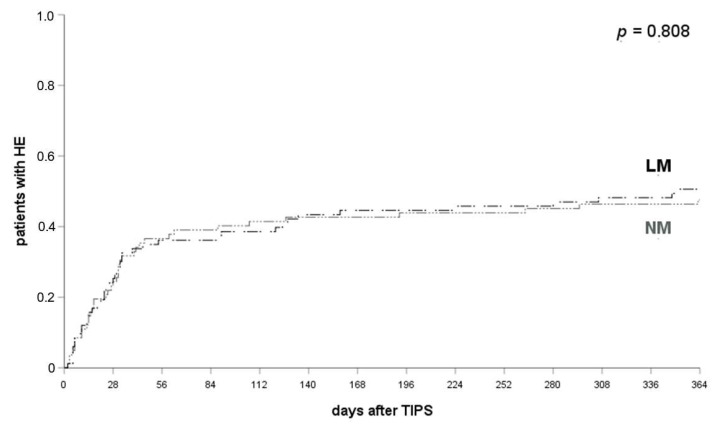
HE occurrence in patients with LM and NM. HE occurred in 38.8% of patients with *NM* and 33.7% of patients with *LM* at 1 month (*p* = 0.581), 44.3% and 40.9% at 3 months (*p* = 0.723) and 53.7% and 50.6% at 12 months after their TIPS implantation (*p* = 0.808). log-rank test applied. HE, hepatic encephalopathy; LM, lactulose monoprophylaxis; NM, no prophylactic medication.

**Figure 2 jcm-10-04763-f002:**
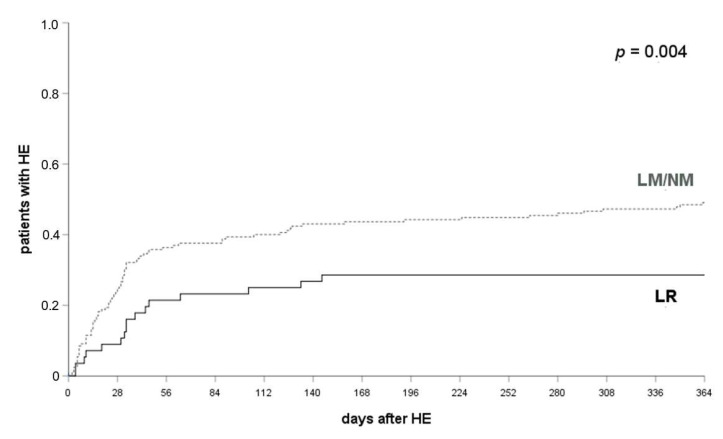
HE occurrence in patients with LR and LR/NM. HE occurred in 36.1% of patients with *LM* or *NM* and 15.8% of patients with *LR* at 1 month (*p* = 0.007), 42.6% and 25.0% at 3 months (*p* < 0.05) and 52.1% and 28.1% at 12 months after their TIPS implantation (*p* = 0.004). A log-rank test was applied. HE, hepatic encephalopathy; LR, lactulose and rifaximin; LM, lactulose monoprophylaxis; NM, no prophylactic medication.

**Figure 3 jcm-10-04763-f003:**
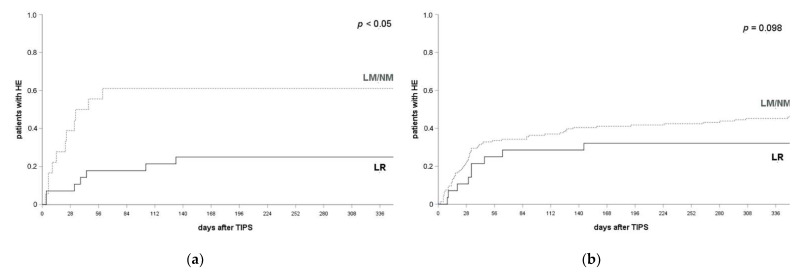
HE occurrence and prophylactic efficacy by history of HE prior to a TIPS implantation. (**a**) In patients with at least one HE episode prior to their TIPS placement, HE occurred in 44.4% of patients with *LM*/*NM* and 10.7% of patients with *LR* at 1 month (*p* = 0.009), 61.4% and 20.0% at 3 months (*p* = 0.003) and 64.7% and 25.0% at 12 months after a TIPS implantation (*p* = 0.007). A log-rank test ws applied. (**b**) In patients with no history of HE prior to their TIPS implantation, HE occurred in 34.7% of patients with *LM*/*NM* and 20.7% of patients with *LR* at 1 month (*p* = 0.173), 39.6% and 29.6% at 3 months (*p* = 0.364) and 50.7% and 31.0% at 12 months after their TIPS implantation (*p* = 0.098). log-rank test applied. HE, hepatic encephalopathy; LR, lactulose and rifaximin; LM, lactulose monoprophylaxis; NM, no prophylactic medication.

**Figure 4 jcm-10-04763-f004:**
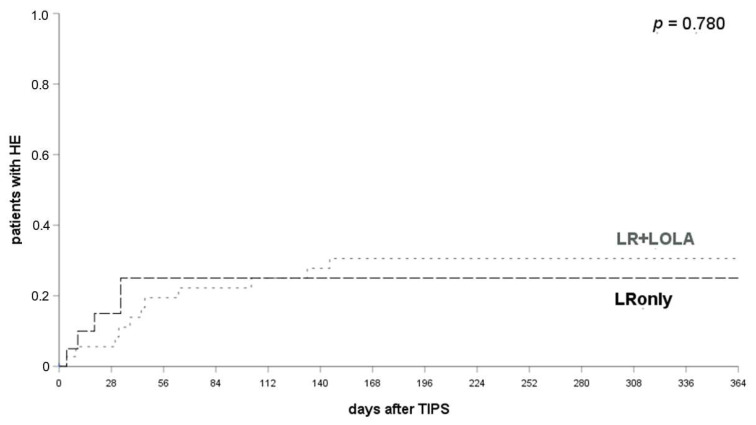
Comparison of HE occurrence in patients with LRonly or LR + LOLA for HE prophylaxis. HE occurred in 20.0% of patients with *LRonly* and 21.6% of patients with *LR + LOLA* at 1 month (*p* = 0.815), 25.0% and 25.0% at 3 months (*p* = 0.979) and 25.0% and 29.7% at 12 months after a TIPS implantation (*p* = 0.780). A log-rank test ws applied. LR, lactulose and rifaximin; LOLA, l-ornithin-l-aspartate; LRonly, LR without LOLA; LR + LOLA, LR with addition of LOLA.

**Table 1 jcm-10-04763-t001:** Baseline characteristics.

Parameter	All Patients % (Total Number) or Median/Mean (SD or Range)	HE Prior TIPS % (Total Number) or Median/Mean (SD or Range)	No HE Prior TIPS % (Total Number) or Median/Mean (SD or Range)	*p* -Value
**n° of patients**	233	21.0 (49)	79.0 (184)	-
sex				0.798
male	60.9 (42)	59.2 (29)	61.4 (113)	
female	39.1 (91)	40.8 (20)	38.6 (71)	
age (median, range, in y)	58 (19–80)	59 (41–77)	58 (19–80)	0.911
PTFE-covered stent	78.1 (182	77.6 (38)	78.2 (144)	0.944
effective stent-diameter	8.0 (6–12)	8.0 (8–12)	8.0 (6–12)	0.984
HE prior TIPS	21.0 (49)	100 (49)	-	-
HE grade				
I	77.6 (38)	77.6 (38)		
II	16.3 (8)	16.3 (8)		
III	6.1 (3)	6.1 (3)		
IV	-	-		
HE prophylaxis				
NM	35.6 (83)	16.3 (8)	40.8 (75)	0.002
LM	2.1 (5)	20.4 (10)	40.8 (75)	0.298
RM	36.5 (85)	4.1 (2)	1.6 (3)	0.008
LR	25.3 (59)	59.2 (29)	16.3 (30)	<0.001
LR + LOLA	16.3 (38)	42.9 (21)	9.2 (17)	<0.001
LRonly	9.0 (21)	16.3 (8)	7.1 (13)	0.046
etiology of liver disease				
alcoholic	51.5 (120)	55.1 (27)	50.8 (93)	0.594
viral	9.4 (22)	8.2 (4)	9.8 (18)	
NAFLD	8.6 (20)	10.2 (5)	8.2 (15)	
other	30.5 (71)	26.5 (13)	31.1 (57)	
Child–Pugh grade				<0.001
A	27.0 (63)	8.2 (4)	32.1 (59)	
B	61.5 (143)	65.3 (32)	60.3 (111)	
C	11.5 (27)	26.5 (13)	7.6 (14)	
indication for TIPS				
ascites	49.4 (115)	55.1 (27)	47.8 (88)	0.894
variceal bleeding	33.0 (77)	20.4 (10)	36.4 (67)	
secondary prophylaxis	84.4 (65)	80.0 (8)	85.1 (57)	
pre-emptive TIPS	9.1 (7)	10.0 (1)	9.0 (6)	
emergency TIPS	6.5 (5)	10.0 (1)	5.9 (4)	
both	17.6 (41)	24.5 (12)	15.8 (29)	
LTX prior TIPS	4.7 (11)	2.0 (1)	5.4 (10)	0.319
HE after TIPS	54.5 (127)	61.2 (30)	52.7 (97)	0.337
HE grade				
I	60.6 (77)	76.7 (23)	55.7 (54)	0.521
II	19.7 (25)	13.3 (4)	21.6 (21)	
III	14.2 (18)	3.3 (1)	17.5 (17)	
IV	5.5 (7)	6.6 (2)	5.2 (5)	
TIPS revision	3.0 (7)	2.0 (1)	3.3 (6)	0.649
TIPS occlusion	0.9 (2)	-	1.1 (2)	0.461
diabetes	28.8 (67)	28.6 (14)	28.8 (53)	0.940
MELD-score	14 (7.2)	18.9 (6.4)	13.6 (7.0)	<0.001
bilirubin (mg/dL)	1.30 (2.46)	2.0 (4.23)	1.20 (1.31)	<0.001
albumin (g/dL)	3.23 (0.64)	3.16 (0.56)	3.32 (0.66)	0.287
creatinine (mg/dL)	1.33 (0.88)	1.61 (1.27)	1.21 (0.67)	0.009
INR	1.31 (0.34)	1.43 (0.30)	1.28 (0.35)	0.053
platelets (ths/µL)	139 (102)	131 (81)	143 (110)	0.892
hemoglobin (mg/dL)	10.1 (4.8)	9.2 (3.6)	10.4 (2.2)	0.002
PSG (mmHg)	18.0 (5.8)	19.0 (5.8)	16.9 (5.7)	0.044

Abbreviations: HE, hepatic encephalopathy; TIPS, transjugular intrahepatic portosystemic shunt; PTFE, polytetrafluoroethylene; NM, no prophylactic medication; LM, lactulose monoprophylaxis; RM, rifaximin monoprophylaxis; LR, lactulose and rifaximin; LOLA, l-ornithin-l-aspartate; LRonly, LR without LOLA; LR + LOLA, LR with LOLA; NAFLD, non-alcoholic fatty liver disease; LTX, liver transplantation; MELD, model of endstage liver disease; INR, international normalized ratio; PSG, portosystemic pressure gradient.

**Table 2 jcm-10-04763-t002:** Predictors of HE within 12 months after TIPS implantation.

Parameter	ß	SE	HR	95% CI for HR	*p*-Value
Univariate Model					
age	0.035	0.013	1.035	1.010–1.061	0.006
HE prior TIPS	1.446	0.448	4.245	1.765–10.212	0.001
PSG before TIPS	0.033	0.026	1.034	0.983–1.087	0.191
PSG after TIPS	−0.008	0.042	0.992	0.913–1.078	0.847
∆PSG	−0.062	0.034	0.940	0.879–1.005	0.171
bilirubin	0.146	0.140	1.157	0.878–1.523	0.300
INR	−0.948	0.697	0.387	0.099–1.519	0.387
creatinine	0.074	0.246	1.076	0.664–1.744	0.765
hemoglobin	−0.027	0.073	0.973	0.843–1.124	0.711
platelet count	0.002	0.002	1.002	0.998–1.006	0.260
diabetes	0.209	0.157	1.233	0.906–1.678	0.184
indication	0.325	0.233	1.384	0.877–2.185	0.162
etiology of liver disease	−0.026	0.065	0.974	0.857–1.107	0.687
Child–Pugh score	0.118	0.275	1.126	0.657–1.929	0.666
effective stent-dameter	0.136	0.119	1.146	0.907–1.447	0.253
multivariate model					
age	0.038	0.013	1.039	1.013–1.066	0.003
HE prior TIPS	1.307	0.449	3.695	1.531–8.917	0.004

Abbreviations: HE, hepatic encephalopathy; TIPS, transjugular intrahepatic portosystemic shunt; ß, beta coefficient; SE, standard error; CI, confidence interval; HR, hazard ratio; INR, international normalized ratio; PSG, portosystemic pressure gradient; ∆PSG, difference (delta) between PSG before TIPS and PSG after TIPS.

**Table 3 jcm-10-04763-t003:** Logistic regression analysis for HE post TIPS implantation by prophylactic regimen.

HE Prior TIPS				No HE Prior TIPS		
Regimen	OR	95% CI	*p*-Value	OR	95% CI	*p*-Value
1 month post TIPS						
LM	0.171	0.020–1.436	0.153	1.018	0.516–2.011	1.000
LR	0.048	0.006–0.366	0.003	0.490	0.177–1.362	0.234
LRonly	0.057	0.004–0.817	0.041	0.940	0.258–3.430	1.000
LR + LOLA	0.044	0.005–0.400	0.005	0.251	0.053–1.185	0.119
3 months post TIPS						
LM	0.133	0.011–1.611	0.145	1.036	0.530–2.023	1.000
LR	0.042	0.004–0.429	0.003	0.647	0.249–1.678	0.483
LR only	0.024	0.001–0.468	0.010	0.768	0.211–2.793	0.759
LR + LOLA	0.051	0.005–0.563	0.009	0.558	0.162–1.929	0.397
12 months post TIPS implantation						
LM	0.111	0.009–1.309	0.134	1.086	0.566–2.083	0.869
LR	0.056	0.006–0.545	0.006	0.450	0.181–1.121	0.121
LRonly	0.024	0.001–0.468	0.010	0.500	0.138–1.809	0.358
LR + LOLA	0.071	0.007–0.729	0.024	0.417	0.133–1.304	0.177

HE incidence was compared to patients receiving NM for prophylaxis. Abbreviations: HE, hepatic encephalopathy; TIPS, transjugular intrahepatic portosystemic shunt; OR, odds ratio; CI, confidence interval; NM, no prophylactic medication; LM, lactulose monoprophylaxis; LR, lactulose and rifaximin; LOLA, l-ornithin-l-aspartate; LRonly, LR without LOLA; LR + LOLA, LR with LOLA.

## Data Availability

The data presented in this study are available on request from the corresponding author.

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
