# Peer review of "Recurrence of Hepatic Encephalopathy after TIPS: Effective Prophylaxis with Combination of Lactulose and Rifaximin"

_jcm, 2021, doi:10.3390/jcm10204763_

Round 1
Reviewer 1 Report
Seifert and colleagues retrospectively assessed efficacy of prophylactic medication for hepatic encephalopathy (HE) after implantation of transjugular portosystemic shunt (TIPS). The authors concluded that higher age and previous HE are risk factors for post TIPS HE. The authors did identify the combination of lactulose plus rifaximin as the best medical prophylaxis in this cases, compared to no treatment or monotherapy with either rifaximin or lactulose. Additionally, there was no additional benefit of administration on l-ornitin-l-aspartate. The discussed topics are of high interest and evidences are still scarce, still I would like to suggest a few observations that may improve the manuscript:
MAJOR:
- The presence of hepatic encephalopathy prior to TIPS placement should be described more specifically, reporting severity of HE episodes, differentiating patients with overt HE from those with covert HE. Moreover, overt HE should be further classified as provoked by precipitating events or non-precipitated. This distinction is detrimental for the stratification of post TIPS risk of HE
- Similarly, post TIPS HE episodes should be better defined in terms of severity, presence of precipitating factors and if a revision of the TIPS was then necessary
- Table 1 reports the inclusion in the cohort of patients with a previous liver transplant, which were not included in the patients selection criteria section. Where these patients showing recurrence of liver cirrhosis? Which was the indication for TIPS placement?
- Besides the already considered variables, other factors should be investigated for risk factor analysis, such as:
- presence of sarcopenia prior to TIPS placement
- final diameter of the stent should be considered
- the indication for TIPS placement for variceal bleeding should be further subclassified as prophylactic, urgent or pre-emptive
- The author reported that the decision whether prophylactic treatment was administered “according to respective standard operating procedures and available guidelines with variability throughout the observational period”. Is it possible to identify which protocols/guidelines have been used over the years?
MINOR:
- The presence of ongoing HE prophylaxis prior to TIPS placement should be reported
- The authors should state whether patients in group NM (no medication) were initiated on medications in case of post TIPS HE occurrence
- The manuscripts contains a few typos:
- Results section, line 113: “…Only patients no prophylactic regimen for HE after TIPS were included in this analysis..”
- Table 1: “…LTX prior TIPS…”
- Discussion section, line 275: “…Our findings show that a history of at least one HE episode prior to TIPS implantation are at higher risk to develop HE after TIPS…”
- English should be revised
Author Response
Reviewer Report 1:
MAJOR:
- The presence of hepatic encephalopathy prior to TIPS placement should be described more specifically, reporting severity of HE episodes, differentiating patients with overt HE from those with covert HE. Moreover, overt HE should be further classified as provoked by precipitating events or non-precipitated. This distinction is detrimental for the stratification of post TIPS risk of HE
- Thank you for those important comments. Data concerning the severity of HE episodes prior and after TIPS are included in table 1 of the revised manuscript. According to West-Haven Criteria, HE episodes can be classified as covert (minimal HE and grade I HE) and overt HE (grade II-IV). As the diagnosis of minimal HE, i.e. the presence of HE according to clinical test without clinical manifestations, is a subject of controversy concerning diagnostic work-up, we included only HE episodes with clinical features ( HE grade I-IV according to West Haven) in the presented work. The respective additions are included in the revised manuscript. Unfortunately, due to the retrospective character of our work, reliable data concerning precipitating factors cannot be included as discussed in lines 283-294 of the original manuscript
- Similarly, post TIPS HE episodes should be better defined in terms of severity, presence of precipitating factors and if a revision of the TIPS was then necessary
- Concerning severity of HE etc. please see our comments above. We additionally added information concerning the necessity of TIPS reductions or TIPS occlusions for the treatment of post TIPS HE in table 1 of the revised manuscript. These additional data certainly contain important information concerning the treatment of post TIPS HE, thank you for that important suggestion.
- Table 1 reports the inclusion in the cohort of patients with a previous liver transplant, which were not included in the patients selection criteria section. Where these patients showing recurrence of liver cirrhosis? Which was the indication for TIPS placement?
- Patients with a history of liver transplantation were included if recurrence of cirrhosis was present and resulted in the necessity of treatment of portal hypertension due to refractory ascites or variceal bleeding (pre-emptive or urgent TIPS)
- Besides the already considered variables, other factors should be investigated for risk factor analysis, such as:
- presence of sarcopenia prior to TIPS placement
- final diameter of the stent should be considered
- the indication for TIPS placement for variceal bleeding should be further subclassified as prophylactic, urgent or pre-emptive
- The presence of sarcopenia in TIPS patients or cirrhotic patients in general is most certainly a subject of highest interest. We have read the recent works, especially those of Nardelli et al. (Clin Gastroenterol Hepatol 2017) and Benmassaoud (Am J Gastronenterol 2020), with greatest interest. Yet, the retrospective diagnosis of sarcopenia is difficult and remains a subject of discussion, as diagnosis is mostly performed via muscle area analysis in retrospective studies and the respective measurements remain a subject of discussion (Derstine et al, J Nutr Health Aging 2017, Derstine et al, Sci reports 2018, Derstine et al, Sinclai Nutrients 2019, Sci reports 2021). The review of Son et al. (Life 2021) provides a great overview concerning the definition and diagnosis of sarcopenia in chronic liver disease. Due to the complexity of diagnosis of this subject, an inclusion in our current work is not feasible and would go beyond the scope of the presented data.
- The final TIPS stent diameter was included in the baseline characteristics as effective stent diameter. We did not find evidence for an impact on development of HE in this cohort.
- Information concerning the respective TIPS indication (secondary prophylaxis, pre-emptive TIPS or emergency TIPS) are now included in table 1. We did not find any differences between the different cohorts included in our manuscript and the subclassification did not provide further evidence to present an independent risk factor or protective factor. This is possibly due to the small number of pre-emptive (7 patients) and emergency (5 patients) TIPS implantations. To maintain a better overview for all potential readers we did not include the subclassification in the univariate or multivariate analysis but maintained ‘variceal bleeding’ as summarizing parameter
- The author reported that the decision whether prophylactic treatment was administered “according to respective standard operating procedures and available guidelines with variability throughout the observational period”. Is it possible to identify which protocols/guidelines have been used over the years?
- The respective standard operating procedures have been regularly revised and updated based on guidelines of the EASL, AASLD, DGVS and internal patient analysis. The addition of LOLA as part of a primary prophylactic regimen after TIPS was part of an internal SOP.
MINOR:
- The presence of ongoing HE prophylaxis prior to TIPS placement should be reported
- We have analyzed the prophylactic regimen beginning with TIPS implantation. Unfortunately, data concerning possibly ongoing prior HE prophylaxis is not available
- The authors should state whether patients in group NM (no medication) were initiated on medications in case of post TIPS HE occurrence
- all patients with post TIPS HE occurrence received secondary prophylaxis after initial treatment. This information is now included in the revised manuscript. Thank you!
- The manuscripts contains a few typos:
- Results section, line 113: “…Only patients no prophylactic regimen for HE after TIPS were included in this analysis..”
- correction included in the revised manuscript
- Table 1: “…LTX prior TIPS…”
- correction included in the revised manuscript
- Discussion section, line 275: “…Our findings show that a history of at least one HE episode prior to TIPS implantation are at higher risk to develop HE after TIPS…”
- correction included in the revised manuscript
- English should be revised
Reviewer 2 Report
The manuscript presents a study of the prophylactic measures in preventing hepatic encephalopathy after TIPS. The authors included 233 patients receiving TIPS and analyzed the risk factors for HE and the role of different prophylactic treatments. Their results are important as there are no definite guidelines on preventing HE in adults after TIPS.
There are some minor improvements that the authors should make to the manuscript:
- editing: line 68, German, line 113... only patients no prophylactic - something missing; line 218 abbreviated HE could be used.
- Table 1: for PTFE, HE prior TIPS, LTX, HE after TIPS, diabetes, there could be only one of the values - as the other is the difference
- Table 3: I would present the results in two main columns: HE prior TIPS (OR, 95%CI, p-value) and no HE prior TIPS (OR, 95%CI, p-value)
- the authors might present the last paragraph as 5. Conclusions
Author Response
Review Report 2:
Comments and Suggestions for Authors
The manuscript presents a study of the prophylactic measures in preventing hepatic encephalopathy after TIPS. The authors included 233 patients receiving TIPS and analyzed the risk factors for HE and the role of different prophylactic treatments. Their results are important as there are no definite guidelines on preventing HE in adults after TIPS.
There are some minor improvements that the authors should make to the manuscript:
- editing: line 68, German, line 113... only patients no prophylactic - something missing; line 218 abbreviated HE could be used.
- respective corrections are included in the revised manuscript
- Table 1: for PTFE, HE prior TIPS, LTX, HE after TIPS, diabetes, there could be only one of the values - as the other is the difference
- Thank you very much for these suggestions. The respective revision of table 1 significantly improves the clarity of table 1 and has thus been included in the revised manuscript
- Table 3: I would present the results in two main columns: HE prior TIPS (OR, 95%CI, p-value) and no HE prior TIPS (OR, 95%CI, p-value)
- Thank you very much this constructive suggestion. We have reorganized table 3 accordingly.
- the authors might present the last paragraph as 5. Conclusions
- The revised manuscript now includes the last paragraph as a separate parapraph ‘5. Conclusions’. Thank you.
Round 2
Reviewer 1 Report
- The authors now included data about concerning the severity of HE episodes prior and after TIPS. However, the occurrence of chronic post TIPS HE, which is the most feared especially for elective TIPS placement, was not included and the lack of data concerning precipitating factors does not allow a correct stratification of post TIPS risk of HE and assessment of prophylaxis efficacy in this context. Provoked HE episodes should be considered separately for the latter aim.
- Besides this, sample numerosity is too low to assess differences between different prophylactic schedules in terms of efficacy.
- Timings of prophylaxis initiation and treatment duration has not been reported, which should be considered for the assessment of treatment efficacy, together with adherence to prescribed treatment
- In the statistical analysis section, the author state that “Kaplan-Meier curves and the log-rank test were used to analyze occurrence of HE and transplant-free survival” however the latter is not present in the result section.
- The inclusion of patients with grade III pre TIPS HE should be commented according to indication to TIPS
- The final diameter of the stent is now included, however, for a correct reading of the data it should be reported as median value and range.
- Post TIPS portosystemic gradient should also be included in the risk analysis as a risk factor for post TIPS HE
- Patients who needed revision of the TIPS were on prophylactic drugs? This data should be commented
- The authors should acknowledge the lack of data about sarcopenia in their risk analysis, as a possible bias
- The identified applied protocols/guidelines used for prophylaxis should be reported in the manuscript
- The lack of data concerning ongoing HE prophylaxis prior to TIPS placement should be acknowledged
Author Response
Dear Editorial Office of Journal of Clinical Medicine,
Again, we would like to thank you for considering our manuscript entitled ‘Recurrence of hepatic encephalopathy after TIPS: effective prophylaxis with combination of lactulose and rifaximin’ for publication in your renowned journal. We had previously improved our manuscript according to the first review reports. The comments and suggestions of the second review round are again of highest standard and provide substantial input to further improve our manuscript.
We were able to integrate the vast majority of the reviewer’s suggestions into the revised manuscript. Nonetheless, two comments cannot be integrated into the current work. the presence of our sarcopenia cannot be further analyzed in the presented study. As proposed by the reviewers, a discussion of this potential bias has been added to the manuscript. Another potential bias, the lack of data concerning precipitating factors of HE, was already discussed in the initial manuscript. We believe that despite this lack of information, our study presents substantial real-world experience data to help improve the lack of evidence concerning post TIPS hepatic encephalopathy.
According to the reviewer’s suggestions we have added several further information in the revised manuscript: information concerning the development of chronic HE and its therapy as well as the respective patient’s prophylactic medication is now included. Besides, information the time of initiation of the respective prophylactic medication is now pointed out. We have also replenished the risk analysis with data concerning post TIPS-portosystemic pressure gradient and the portosystemic pressure difference before and after TIPS without finding a significant impact on the development of post TIPS HE. Please find ‘point-by-point’ responses to the review reports in italic below. All changes are marked in the submitted document jcm-1385606_manuscript_revision_2. With the help of the reviewers’ comments, further important improvements to our manuscript could be achieved. We sincerely hope that our work fulfills the high standards of your renowned journal. There is scarce evidence concerning prophylaxis of HE post TIPS and we believe that our work can contribute to the development of a better patient management.
In case of further questions or suggestions, please to not hesitate to contact us.
On behalf of all authors,
Sincerely,
Moritz Wildgruber, MD, PhD,
Corresponding author
Department of Radiology, University Hospital LMU Munich, 81377 Munich, Germany
E-Mail: Moritz.Wildgruber@med.uni-muenchen.de
Reviewer Report Round 2:
- The authors now included data about concerning the severity of HE episodes prior and after TIPS. However, the occurrence of chronic post TIPS HE, which is the most feared especially for elective TIPS placement, was not included and the lack of data concerning precipitating factors does not allow a correct stratification of post TIPS risk of HE and assessment of prophylaxis efficacy in this context. Provoked HE episodes should be considered separately for the latter aim.
- Additional to the inclusion of data concerning the severity of HE, data concerning chronic post TIPS HE are now included in the further revised manuscript (see results, 3.8. Development of chronic HE). Chronic HE (defined as HE refractory to pharmaceutical intervention) was present in 9 patients and treated with TIPS revision (7/9 patients) or occlusion (2/9 patients). We agree that especially chronic HE post TIPS is of utmost interest and the addition of this data is important to complete the presented data. Thank you very much for this comment.
- We agree that the lack of data concerning precipitating factors hampers the conclusive power of our work as pointed out in the discussion part (lines 321-327). It has to be acknowledged that in a real world setting and clinical reality often the exact trigger/precipitating factor cannot be identified. This represents an important burden concerning all retrospective studies on the subject of hepatic encephalopathy. Nonetheless we believe that the presented data implies important findings from a large real world cohort.
- Besides this, sample numerosity is too low to assess differences between different prophylactic schedules in terms of efficacy.
- It is out of question that a bigger patient cohort would be desirable to draw clinical implications from our work. We believe that retrospective clinical studies like the presented work have to be seen, as pointed out in the discussion, as important contributions for the encouragement of prospective studies. Nonetheless, the presented results present important significant data from our point of view. In a field with sparse scientific evidence, our retrospective work may contribute to the development more scientific based evidence and recommendations in the future.
- Timings of prophylaxis initiation and treatment duration has not been reported, which should be considered for the assessment of treatment efficacy, together with adherence to prescribed treatment
- We have now included timing and treatment duration in the revised manuscript (lines 81-84). Adherence to treatment has been controlled at regular ambulatory visits at 3, 6 and 12 months after TIPS implantation. Further control was not performed. The specifications and included in the revised manuscript. Thank you for this important suggestion.
- In the statistical analysis section, the author state that “Kaplan-Meier curves and the log-rank test were used to analyze occurrence of HE and transplant-free survival” however the latter is not present in the result section.
- We have corrected the respective sentence in the revised manuscript, thank you!
- The inclusion of patients with grade III pre TIPS HE should be commented according to indication to TIPS
- We have added a comment concerning the 3 patients with grade III HE in their history prior to TIPS. ‘Of the 3 patients with grade III HE in the past, two patients had refractory variceal bleeding and one patient had refractory ascites as TIPS indication. All grade III HE episodes had occurred more than 12 weeks prior to TIPS implantation. No symptoms were present at the time of TIPS implantation.‘ (see lines 151-154). We are aware of the fact that a grade III HE represents a contraindication to TIPS if present at the time of (planned) TIPS implantation according to most available guidelines. Grade III HE was not present in those patients at the time of TIPS placement. Nonetheless, because of their prior history with severe HE an extensive interdisciplinary discussion was performed in the decision-making concerning TIPS implantation. We agree that it is of importance to clarify the exact circumstances at the time of TIPS implantation for potential readers. We would like to thank the reviewers of this important suggestion.
- The final diameter of the stent is now included, however, for a correct reading of the data it should be reported as median value and range.
- We have corrected the data presentation accordingly, thank you! (see table 2)
- Post TIPS portosystemic gradient should also be included in the risk analysis as a risk factor for post TIPS HE
- We have included both the post TIPS portosystemic pressure gradient as well as the pre- and post-TIPS pressure difference in the risk analysis (see table 2). A significant impact on the development of post TIPS HE was not observed
- Patients who needed revision of the TIPS were on prophylactic drugs? This data should be commented
- This is a very important point, thank you draw our attention on this subject. We have included additional data concerning prophylactic medication in the respective patients in the revised manuscript
- The authors should acknowledge the lack of data about sarcopenia in their risk analysis, as a possible bias
- Thank you for bringing up this important subject. We did discuss this subject in response to the first review reports (see below). Now, we have included an acknowledgement of this potential bias the revised manuscript citing the work of Nardelli et al. and Benmassaoud et al. (see 328-334).
- The identified applied protocols/guidelines used for prophylaxis should be reported in the manuscript
- Thank for this suggestion, we have included a report concerning the guidelines and internal SOPs in the re-revised manuscript (lines 98-102)
- The lack of data concerning ongoing HE prophylaxis prior to TIPS placement should be acknowledged
- Thank you for putting the focus on this subject. We have included an acknowledgement of this limitation both in “2. Material and Methods” as well as in the discussion.